# Cross-links Matter for Link Prediction: Rethinking the Debiased GNN from a Data Perspective

**Zihan Luo**[1], **Hong Huang**[1]*, **Jianxun Lian**[2], **Xiran Song**[1], **Xing Xie**[2], **Hai Jin**[1]

[1] National Engineering Research Center for Big Data Technology and System,
Service Computing Technology and Systems Laboratory,
Cluster and Grid Computing Lab,
School of Computer Science and Technology,
Huazhong University of Science and Technology, Wuhan, China
[2] Microsoft Research Asia, Beijing, China
{zihanluo,honghuang,xiransong,hjin}@hust.edu.cn,
{jianxun.lian,xingx}@microsoft.com

## Abstract

Recently, the bias-related issues in GNN-based link prediction have raised widely spread concerns. In this paper, we emphasize the bias on links across different node clusters, which we call cross-links, after considering its significance in both easing information cocoons and preserving graph connectivity. Instead of following the objective-oriented mechanism in prior works with compromised utility, we empirically find that existing GNN models face severe data bias between internal-links (links within the same cluster) and cross-links, and this inspires us to rethink the bias issue on cross-links from a data perspective. Specifically, we design a simple yet effective twin-structure framework, which can be easily applied to most GNNs to mitigate the bias as well as boost their utility in an end-to-end manner. The basic idea is to generate debiased node embeddings as demonstrations and fuse them into the embeddings of original GNNs. In particular, we learn debiased node embeddings with the help of augmented supervision signals, and a novel dynamic training strategy is designed to effectively fuse debiased node embeddings with the original node embeddings. Experiments on three datasets with six common GNNs show that our framework can not only alleviate the bias between internal-links and cross-links but also boost the overall accuracy. Comparisons with other state-of-the-art methods also verify the superiority of our method.

## 1 Introduction

Recently, due to the strong capability in learning the latent representation of graph structure data, *Graph Neural Networks* (GNNs)-based link prediction methods have received increasing research interests and show excellent performance in recommendation systems [33, 34], bioinformatics [7, 16], and knowledge graph [24, 40]. However, these link prediction methods often prioritize performance without considering potential bias on the sensitive attributes of nodes, such as genders or regions, thus leading to social risks or the creation of informa-

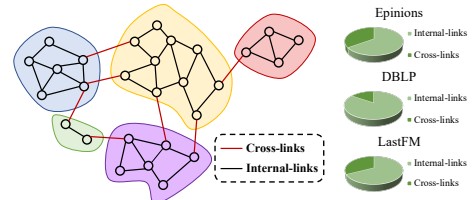

Figure 1: A toy graph for illustrating the concept of topological communities. The proportions of two kinds of links also are provided on the right.

---

*Hong Huang is the corresponding author.

37th Conference on Neural Information Processing Systems (NeurIPS 2023).

tion cocoons. For instance, in social recommendations, models tend to recommend users with friends belonging to the same region, limiting users' opportunities to connect with the outside world [46]. Similarly, in political news recommendations, recommenders prefer to deliver content that aligns with users' partisan beliefs, filtering out different perspectives and narrowing users' scopes [23]. To tackle these issues, researchers have put forward several solutions. For example, FLIP [28] employs adversarial learning to encourage the model to predict outcomes that are independent of sensitive attributes. Similarly, CFC [4] aims to debias the node embeddings with compositional adversarial constraints. FairAdj [20] learns edge weights to generate a fair adjacency matrix, which is used for subsequent link prediction tasks. UGE [38] explores unbiased node embeddings from an unobserved graph, and proposes two kinds of methods based on reweighting and regularization constraints.

Although these methods make great progress in mitigating the bias in link prediction, there are two issues remaining unsolved: 1) All these methods generally modify the objective functions, like adding regularization constraints [4, 28] or reweighting [38] for mitigating the bias. However, the mechanism of simply modifying objective function may influence the optimization trajectory of the model and result in convergence towards a sub-optimal status. The experimental trade-off between debias and utility on multiple prior works [4, 20, 28, 38] also supports this claim. 2) Existing methods simply focus on sensitive attributes of nodes while ignoring the bias based on graph topology. In fact, due to GNNs' heavy dependence on the aggregation of neighborhood messages,

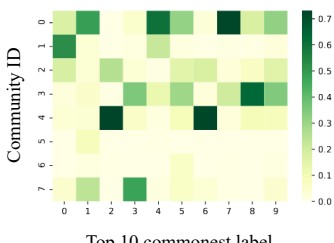

Figure 2: Distributions for top 10 labels in 10 different communities detected by Louvain on LastFM

for the link prediction tasks, GNNs tend to wire new links inside the local community[2] (denoted as **internal-links**), while ignoring the links connecting different communities (denoted as **cross-links**), and this kind of bias may leave the graph in danger of being trapped in information cocoons or disconnected [21, 28], as shown in Figure 1. Empirical evidence can also be provided in Figure 2, in which we illustrate the top 10 commonest labels' distributions in different communities. It can be seen that each community contains its own specific information pattern [26], making it challenging for one single community to encompass all diverse information in a network. This observation implies the propensity of graphs to form information cocoons with insufficient cross-links as bridges[3]. Motivated by these findings, we aim to design a GNN model that could boost the link prediction performance on cross-links and mitigate the bias between internal-links and cross-links without sacrificing utility.

By rethinking the problem of bias between internal-links and cross-links with a fresh insight from the data perspective, we statistics the proportion of internal-links and cross-links in three real-world datasets in Figure 1, and we can find that the number of internal-links far exceeds that of cross-links. This kind of data bias may be further enlarged and perpetuated by stacking GNN aggregation operations and finally leads to a biased link prediction. In light of this, we propose a simple yet effective twin-structure framework, which consists of two independent GNN models to mitigate the bias between internal-links and cross-links. Specifically, we first divide the whole graph into multiple communities that share similar topological locations and differentiate links into internal-links and cross-links. After that, in order to alleviate the data bias between internal-links and cross-links, supervision augmentation based on multiple rules is proposed to increase the supervised signals for cross-links, which could help the GNNs to better capture the patterns of cross-links and further generate debiased node embeddings. Subsequently, to avoid utility degradation, we design an embedding fusion module to merge original node embeddings and debiased node embeddings with a dynamic training strategy. In this way, the embedding fusion module could effectively preserve the performance of internal-links while alleviating the bias between internal-links and cross-links. Our main contributions can be summarized as follows:

- We reveal a significant link prediction bias based on the graph topology, i.e. the bias between internal-links and cross-links, and design a model-agnostic framework, which can help most of the GNNs address such bias without compromising utility.

---

[2]Generally, it refers to a cluster of nodes that enjoy similar topological location, which can be detected by community detection algorithms. In this paper, we deploy the Louvain algorithm [2] as the default detection method.

[3]Further analysis on elaborating the role of cross-links is provided in Section A of the Appendix.

- We propose three core components in our framework, including supervision augmentation, twins-GNN, and embedding fusion module. The combination of supervision augmentation and twins-GNN can generate debiased embeddings from a data aspect as demonstrations, while the embedding fusion module can effectively filter out the noise and preserve the utility of GNN.
- We evaluate our method on three real-world datasets with six common GNNs as base models and as well as compare them with other state-of-the-art baselines. The experimental results consistently demonstrate that our framework could effectively reduce the bias between internal-links and cross-links with even improved overall performance.

## 2 Related Work

**Link Prediction**   As one of the most important applications in graph learning, link prediction algorithms have received extensive attention in the last two decades [3, 12, 14, 29]. The traditional link prediction methods are mainly based on some heuristic metrics on the graph structure, such as CN [25], AA [1], and Jaccard coefficient. In recent years, shallow embedding-based link prediction methods such as DeepWalk [29], Node2Vec [12], and LINE [36] have also emerged. With the powerful capability in representation learning on graph structure data, GNNs have achieved great progress in link prediction tasks as well [3, 13, 14, 19, 37, 48]. For example, LightGCN discovers the redundancy of non-linear activation and feature transformation and achieves better performance on recommendations [14]. PPRGo [3] uses a Personalized Page Rank matrix to efficiently approximate the multi-hop aggregation, which breaks the classical message-passing paradigm of GNNs. Following LightGCN, UltraGCN [27] approximates the infinite-layer aggregation for better recommendations. Besides, instead of learning from the whole graph, another line of research focuses on learning from relatively small subgraphs, such as SEAL [44] and SUREL+ [42], which also achieve remarkable performance in link prediction.

**Bias Related Issues in GNNs**   Recently, bias-related issues have been widely concerned in graph neural networks, termed as "debias" or "fairness". Concentrating on node classification tasks, there is a branch of works on achieving fair results that are uncorrelated with sensitive attributes, including FairGNN [8], FairVGNN [39], EDITS [9], and EqGNN [32]. As for bias in link prediction, based on node2vec [12], Fairwalk [30] proposes a more fair random walk strategy, but fails to avoid the decrease in models' utility after achieving satisfactory fairness. Both FLIP [28] and CFC [4] try to mitigate the dyadic bias by proposing adversarial constraints in the loss function. Li et al. [20] designed a model named FairAdj, which could generate a fair adjacency matrix with different edge weights to address the dyadic bias with competitive utility. Similar to FairAdj, UGE [38] derives unbiased embeddings from an unobserved graph that involves no sensitive attribute information. However, all these methods modify the original objective functions and face a trade-off between debias and utility.

## 3 Methodologies

### 3.1 Problem Formulation

In this work, a graph is denoted by $\mathcal{G} = (\mathcal{V}, \mathcal{E})$, and it consists of $N = |\mathcal{V}|$ vertices and $|\mathcal{E}|$ links. In the real world, nodes on the graph spontaneously form local communities [26], such as social circles in social networks, and various bundles of also-buy items on product graphs. We use $\mathcal{C}(v) \in \mathbb{N}$ to denote the community membership of a given node $v$ and define *cross-links* and *internal-links*:

**Definition 1.   *Cross-links and Internal-links.** Given a link $\langle u, v \rangle$, it belongs to cross-links $\mathcal{E}_{cr}$ if its endpoints' memberships satisfy $\mathcal{C}(u) \neq \mathcal{C}(v)$; otherwise, it belongs to internal-links $\mathcal{E}_{in}$ as it satisfies $\mathcal{C}(u) = \mathcal{C}(v)$. So according to the endpoints' memberships, the links $\mathcal{E}$ in a graph could be divided into $\mathcal{E}_{in}$ and $\mathcal{E}_{cr}$ two sets exclusively, i.e. $\mathcal{E}_{in} \cap \mathcal{E}_{cr} = \varnothing$ and $\mathcal{E}_{in} \cup \mathcal{E}_{cr} = \mathcal{E}$.*

Centering on the link prediction based on GNNs, each node will be mapped to an $D$ dimensional embedding vector $\mathbf{z} \in \mathbb{R}^D$ by a GNN encoder. The dot product score of a pair of nodes' embeddings implies the GNN model's confidence on whether there will be a potential link between two given vertices. Our goal is to mitigate the bias between internal-links and cross-links, i.e.:

**Definition 2.   *Bias between internal-links and cross-links (Bias).** Given a trained GNN model, a confidence matrix $\mathbf{P} \in \mathbb{R}^{N \times N}$ can be inferred by calculating the dot product of endpoints'*

*embeddings, and a higher $\mathbf{P}_{uv}$ indicates that model predicts that it very likely exists a link between node $u$ and node $v$. Let the notation $\mathcal{M}$ represent an evaluation metric in link prediction tasks, and we use the difference between two kinds of links' average performance to define the bias:*

$$Bias = |\mathbb{E}[\mathcal{M}(\mathbf{P})|\mathcal{E} = \mathcal{E}_{in}] - \mathbb{E}[\mathcal{M}(\mathbf{P})|\mathcal{E} = \mathcal{E}_{cr}]| \tag{1}$$

**Task definition.** To address such bias, a GNN recommender should balance the performance on internal-links and cross-links, so that the final model could show equivalent capability on mining true internal-links and cross-links. The goal of this paper is to design a model-agnostic framework, which can enhance the embedding ability of a given GNN model so that the bias metrics are reduced while the accuracy metrics are at least not decreased:

$$Bias(\mathcal{M}(\mathbf{P}|\mathbf{Z})) < Bias(\mathcal{M}(\mathbf{P}|\mathbf{Z}')), \quad \mathcal{M}(\mathbf{P}|\mathbf{Z}) \geq \mathcal{M}(\mathbf{P}|\mathbf{Z}') \tag{2}$$

where $\mathbf{Z}'$ represents the original node embeddings given by a GNN, and $\mathbf{Z}$ represents the node embeddings enhanced by our framework.

## 3.2 Framework Overview

We propose a twin-structure GNN-enhancement framework, which can reduce the bias between cross-links and internal-links without hurting the overall performance. An overview of our framework is illustrated in Figure 3. Its key components include supervision augmentation, Twins GNNs, and an embedding fusion module. Details will be provided in the following sections. In traditional graph embedding models, existing links on the graph are used as supervision signals. In our framework, we first generate a certain number of pseudo cross-links and form the augmented supervision signals. Then, we use two GNNs with the same architecture to model the two sets of supervision signals (original supervision and

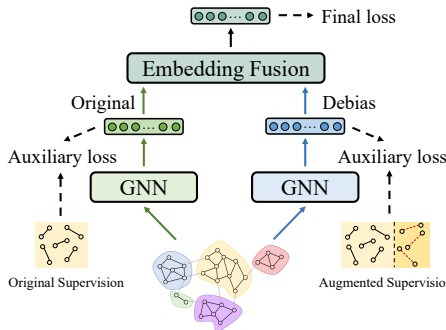

Figure 3: The overview of our framework

augmented supervision) independently, leading to original node embeddings and debiased node embeddings. Lastly, to avoid the impact of noise during supervision augmentation, an embedding fusion module is proposed, which ensures that the final embeddings will retain the merits of both original node embeddings and debiased node embeddings.

## 3.3 Supervision Augmentation

As revealed in Figure 1, in most datasets, usually the vast majority of links on the graph are internal-links, and this kind of disparity may cause the learned model to be biased to internal-links while neglecting the performance of the cross-links in order to obtain better overall metrics. Therefore, we design an augmentation step to alleviate the supervision signal sparsity issue.

**Jaccard based augmentation.** Inspired from the concept of edge strength [11], a set of $K$ pairs of nodes $\{\langle \hat{u}_1, \hat{v}_1 \rangle, ..., \langle \hat{u}_K, \hat{v}_K \rangle\}$, which satisfy $\mathcal{C}(\hat{u}_i) \neq \mathcal{C}(\hat{v}_i), \quad \forall i \in \{1, ..., K\}$, and have the top $K$ highest Jaccard coefficient score $\mathcal{J}(\hat{u}, \hat{v})$, are wired as pseudo cross-links. Specifically, the $\mathcal{J}(\hat{u}, \hat{v})$ can be formulated as:

$$\mathcal{J}(\hat{u}, \hat{v}) = \frac{|\mathcal{N}(\hat{u}) \cap \mathcal{N}(\hat{v})|}{|\mathcal{N}(\hat{u}) \cup \mathcal{N}(\hat{v})|} \tag{3}$$

and $K = k * (|\mathcal{E}_{in}| - |\mathcal{E}_{cr}|)$, where $k$ is a hyper-parameter to control the supervision augmentation size $K$, and $\mathcal{N}(\cdot)$ represents the neighbor nodes of a given node. $\mathcal{J}(\cdot, \cdot)$ measures the proportion of common neighbors between two nodes, and intuitively, a pair of nodes with more common neighbors are more likely to form a potential link, which ensures the high confidence of augmented signals.

**Random walk based augmentation.** However, when a network is extremely dense, adopting Jaccard-based augmentation will consume too much computing resource due to the "neighbor explosion" issue. What's more, the Jaccard-based supervision augmentation may easily only cover nodes around the border of communities, and nodes in the center of a community may still fail to benefit from the

augmentation. To solve these problems, we propose a new supervision augmentation method based on random walks. Specifically, we take random walks in the original graph, and each pair of nodes $\langle \hat{u}, \hat{v} \rangle$ that appear in the same walk path will be recorded. Next, cross-community node pairs, *i.e.,* $\mathcal{C}(\hat{u}) \neq \mathcal{C}(\hat{v})$, with the top $K$ highest co-occurrence frequency will be screened out as augmented supervision signals. Similarly, $K = k * (|\mathcal{E}_{in}| - |\mathcal{E}_{cr}|)$ represents the supervision augmentation size.

Note that, we only generate pseudo supervision signals, and the original graph structure, which is used for message passing in GNNs, remains unchanged. We believe that the lack of supervision plays a much more important role in causing the bias between internal-links and cross-links and adding supervision signals is a more straightforward way to alleviate this kind of bias. Let $\mathcal{E}^P$ denote the newly constructed edge set. The original supervision signals, which are the original links on the graph, are denoted by $\mathcal{E}^O$, and the final augmented supervision signals are denoted by $\mathcal{E}^A = \mathcal{E}^O \cup \mathcal{E}^P$.

## 3.4 Twins GNNs

Recently, the retrieval-based learning paradigm, in which the core idea is to provide supporting examples that can be used as references for the model, has extracted increasing research interest in the field of natural language processing [35]. Inspired by this, it would be a promising way to generate debiased node embeddings as references for our framework to mitigate the bias between internal-links and cross-links. Specifically, we let $\mathcal{E}^O$ and $\mathcal{E}^A$ to guide the training of two twin GNNs, respectively. The two twin GNNs share the same model architecture but have independent sets of parameters. To this end, the twin-structure GNNs could generate two kinds of embeddings named original node embeddings $\mathbf{Z}^O$ and debiased node embeddings $\mathbf{Z}^A$, respectively. In order to make sure that the twin-structure GNNs could precisely generate original embeddings and debiased embeddings for further embedding fusion, we follow the literature [14, 34] and employ one of the most classic loss functions in recommendation systems - BPRLoss to design our two auxiliary loss functions for the twin-structure GNNs, which can be generalized as:

$$\mathcal{L} = -\frac{1}{|\mathcal{E}|} \sum_{\langle u,v \rangle \in \mathcal{E}} \log \sigma(r_{u,v} - r_{u,\hat{v}}) \tag{4}$$

where $\mathcal{E}$ could be $\mathcal{E}^O$ or $\mathcal{E}^A$ depending on the type of generated embeddings, and the auxiliary loss functions, are denoted as $\mathcal{L}^O$ and $\mathcal{L}^A$, respectively. $\sigma$ is an activation function like sigmoid. $\langle u, \hat{v} \rangle$ denotes a negative sample, and $r_{u,v}$ denotes the prediction score of a given node pair $\langle u, v \rangle$. Note that, other alternative loss functions with multiple negative samples can be also easily deployed to our method, which is determined by the practitioners according to their practical applications. Intuitively, with the balanced proportion of two types of links in $\mathcal{E}^A$ after supervision augmentation, the debiased node embeddings $\mathbf{Z}^A$ could show excellent capability in eliminating the bias between internal-links and cross-links. The debiased node embeddings $\mathbf{Z}^A$ will become a good demonstration for subsequent embedding fusion modules in the further learning of final embeddings.

## 3.5 Embedding Fusion

Although we set several heuristic rules in Section 3.3 to ensure that the generated signals have high confidence, the supervision augmentation stage will inevitably introduce noise. Thus, $\mathbf{Z}^A$ is trained with impure supervision. If fusing $\mathbf{Z}^A$ into $\mathbf{Z}^O$ in a naive way, such as a simple averaging, although the performance on the cross-links can be improved, the internal-links' will be considerably degraded.

Thus, we adopt an embedding fusion module $F(\cdot)$ that merges two kinds of embeddings, i.e. original node embeddings $\mathbf{Z}^O$ and debiased node embeddings $\mathbf{Z}^A$, to refine meaningful information from the debiased embeddings and fuse them into the original embeddings:

$$\mathbf{H}_i = \mathrm{CONCAT}(\mathbf{Z}_i^O, \mathbf{Z}_i^A) \tag{5}$$

$$\mathbf{Z}_i = F(\mathbf{H}_i, \theta^F) = \mathbf{W}_2 \, \sigma(\mathbf{W}_1 \mathbf{H}_i + \mathbf{b}_1) + \mathbf{b_2} \tag{6}$$

where $\mathbf{W}_1 \in \mathbb{R}^{D \times 2D}$, $\mathbf{b}_1 \in \mathbb{R}^D$, $\mathbf{W}_2 \in \mathbb{R}^{D \times D}$, $\mathbf{b}_2 \in \mathbb{R}^D$ are the trainable parameters of $F(\cdot)$, and can be abbreviated as $\theta^F$ for conciseness. $\sigma$ is the activation function and $D$ is the embedding dimension. Note that, both $\mathbf{Z}_i^O$ and $\mathbf{Z}_i^A$ are output embeddings from GNN models supervised by auxiliary loss, and we believe that there is no need to further deploy a complex network during embedding fusion, hence we simply show a 2-layer MLP in Eq.(6).

As the core component of our framework, we argue that the final embeddings $\mathbf{Z}$ after embedding fusion function $F(\cdot)$ should filter signal noise in $\mathcal{E}^A$ while preserving as much correct augmented information as possible. Therefore, we optimize embedding fusion module $F(\cdot)$ and twin-GNNs by minimizing the following objective function :

$$\mathcal{L}^F = -\frac{1}{|\mathcal{E}^O|} \sum_{\langle u,v \rangle \in \mathcal{E}^O} \log \sigma(r_{u,v} - r_{u,\hat{v}}) \tag{7}$$

---

**Algorithm 1** Proposed training process

---

**Input:** Graph $\mathcal{G}$ with link set $\mathcal{E}^O$. Hyper-parameters: $\alpha$, $\beta$, and $T$, learning rate $\gamma^O$, $\gamma^A$.
**Output:** Twin GNNs' parameters $\theta^O, \theta^A$, and embedding fusion module parameters $\theta^F$
 1: Randomly initialize twins GNN models with $\theta^O, \theta^A$, embedding fusion module with $\theta^F$.
 2: Split $\mathcal{G}$ into $|\mathcal{C}|$ communities and categorize links into internal-links and cross-links.
 3: Select augmented supervision signals $\mathcal{E}^A$ by the Jaccard coefficient or co-occurrence frequency.
 4: **while** not converged **do**
 5:     Compute $\mathcal{L}^O$ and $\mathcal{L}^A$ by Eq.(4)
 6:     Update twins GNN models: $\theta^O \leftarrow \theta^O + \gamma^O \cdot \nabla_{\theta^O} \mathcal{L}^O, \quad \theta^A \leftarrow \theta^A + \gamma^A \cdot \nabla_{\theta^A} \mathcal{L}^A$
 7:     Compute learning rate $\gamma_t^F$ and step size $S_t$ by Eq.(8)
 8:     **for** $step = 1$ **to** $S_t$ **do**
 9:         Compute $\mathcal{L}^F$ by Eq.(7)
10:         Update embedding fusion module: $\theta^F \leftarrow \theta^F + \gamma_t^F \cdot \nabla_{\theta^F} \mathcal{L}^F$
11:         Update twins GNN models: $\theta^O \leftarrow \theta^O + \gamma_t^F \cdot \nabla_{\theta^O} \mathcal{L}^F, \quad \theta^A \leftarrow \theta^A + \gamma_t^F \cdot \nabla_{\theta^A} \mathcal{L}^F$
12:     **end for**
13: **end while**
14: **return** $\theta^O, \theta^A, \theta^F$

---

**Algorithm 2** Proposed inference process

---

**Input:** Graph $\mathcal{G}$ with link set $\mathcal{E}^O$. Trained Twin GNNs' parameters $\theta^O, \theta^A$, and embedding fusion module parameters $\theta^F$
 1: Get node embeddings $Z^O$ through parameters $\theta^O$
 2: Get node embeddings $Z^A$ through parameters $\theta^A$
 3: Get node embeddings $Z$ through combining $Z^O$ and $Z^A$ with embedding fusion module $\theta^F$
 4: Evaluation through test node pairs with node embeddings $Z$

---

### 3.6 Implementation Algorithms

For a better understanding of our method, here we summarize the detailed training process and inference process, and the pseudo codes are provided in Algorithm 1 and Algorithm 2, respectively.

***Training process***: Firstly, during the preprocessing stage, $C = |\mathcal{C}|$ communities are detected through Louvain algorithm [2] and all links are categorized into internal-links or cross-links, respectively (line 2). In addition, newly constructed supervision signals $\mathcal{E}^P$ are also generated during supervision augmentation (line 3). After that, original node embeddings $\mathbf{Z}^O$ and debiased node embeddings $\mathbf{Z}^A$ are generated by minimizing the auxiliary loss functions in Section 3.4 (lines 5-6) with learning rate $\gamma^O$ and $\gamma^A$, respectively. The two sets of generated embeddings will be forwarded to the embedding fusion module $F(\cdot)$ to get the final embeddings $\mathbf{Z} = F(\mathbf{Z}^O, \mathbf{Z}^A)$, and minimize the final loss function with multiple training steps (lines 8-12.) Essentially, the embedding fusion module $F(\cdot)$ is highly related to the quality of original embeddings $\mathbf{Z}^O$ and debiased embeddings $\mathbf{Z}^A$, thus we introduce a dynamic training strategy to avoid excessive training for embedding fusion module when $\mathbf{Z}^O$ and $\mathbf{Z}^A$ have not been stable yet. Specifically, the learning rate $\gamma_t^F$ and training step $S_t$ for $F(\cdot)$ at epoch $t$ is:

$$\gamma_t^F = \alpha * \frac{1}{1 + e^{-(t-T)}} \quad , \qquad S_t = \beta * \frac{1}{1 + e^{-(t-T)}} \tag{8}$$

where $\alpha$, $\beta$, and $T$ are hyper-parameters to control the learning rate, training steps, and the epoch that $F(\cdot)$ can believe the quality of input embeddings have been stable, respectively.

*Inference process*: During the inference stage, we use the node embeddings $\mathbf{Z}$ for evaluation (line 4), which are the output of the embedding fusion module. Specifically, we first get node embeddings $\mathbf{Z}^O$ and $\mathbf{Z}^A$ through the trained twins GNN in advance (lines 1-2), and then perform embedding fusion operation (line 3) to get the final node embeddings $\mathbf{Z}$.

## 4 Experiment

### 4.1 Experimental Settings

**Datasets.** We conduct our experiments on three real-world networks from SNAP[4] and RecBole[5] [47] public datasets. Specifically, to verify our framework's extendibility, we choose two kinds of networks according to the types of interaction between nodes, including user-user (Epinions, DBLP), and user-item (LastFM). **Epinions** is a who-trust-whom social network extracted from

Table 1: Dataset Statistics

| Datasets | Users | Items | Interactions | Density | Type |
|---|---|---|---|---|---|
| Epinions | 75,879 | - | 508,837 | 0.000088 | User-user |
| DBLP | 317,080 | - | 1,049,866 | 0.000010 | User-user |
| LastFM | 1,892 | 17,632 | 92,834 | 0.002783 | User-item |

the online review site Epinions.com. Each node represents a consumer, and each directed link represents a consumer-to-consumer trust connection. **DBLP** is a co-author social network, which is collected from the DBLP computer science bibliography, and two author nodes are connected if they used to publish at least one paper together. **LastFM** dataset contains users' listening information from the Last.fm online music system and each listening event represents a user-artist interaction. The detailed statistics for these three datasets are reported in Table 1, and other experimental settings on datasets are described in Section B in the Appendix.

**Base models.** Because our proposed method is a model-agnostic framework, which is compatible with almost all GNN-based link prediction models, here we choose six common and effective GNNs as our base models, including **GraphSAGE** [13], **GIN** [41], **GAT** [37], **LightGCN** [14], **PPRGo** [3], and **UltraGCN** [27]. We verify if our framework could mitigate the bias between internal-links and cross-links without hurting the original overall performance.

**Reproducibility.** All experiments are run on machines with the same configuration: Intel(R) Xeon(R) CPU E5-2680 and Tesla V100 GPU with 32GB memory. How to detect the communities of a network is not the research focus of this paper, and here we apply the Louvain algorithm [2] to detect the communities in a graph after considering its high speed and effectiveness. Other alternative community detection algorithms and hyper-parameters will be discussed in the Appendix. The source code and data are available at `https://github.com/CGCL-codes/Cross-links-Bias`.

### 4.2 Main Results

In this part, we show the performance of our method in terms of both link prediction utility and debias effectiveness after applying it to various GNN models.

The main results on three datasets are reported in Table 2. For Epinions and DBLP, we use Jaccard-based augmentation here, and considering the high density of LastFM, we use random walk-based augmentation. More results with random walk-based augmentation on Epinions and DBLP are listed in Section C in the Appendix for reference. The internal-links performance (Internal.), cross-links performance (Cross.), and overall performance (Overall) are calculated on internal-links in the test set, the cross-links in the test set, and the whole test set respectively. To verify our method's capability of addressing the bias, $Bias$ in Eq.(1) is also computed to evaluate the performance difference between internal-links and cross-links. Table 2 shows the results of our approach and base models, and we can have the following observations:

- The performance of internal-links outperforms that of cross-links under all settings, which indicates the link prediction bias between internal-links and cross-links is widespread. Note that, after proposing our framework, we can only ease this kind of bias to some extent, which inspires us the reasons for the poor performance on cross-links may not only come from the data perspective, and we leave this problem as a future work.

---

[4]`https://snap.stanford.edu/data/index.html`
[5]`https://recbole.io/index.html`

Table 2: Link prediction performance (Hits@50) of internal-links, cross-links, and the whole link set of our methods and corresponding base models on three real-world datasets. The results are reported in percentage (%). We **bold** the results when our framework improves the base GNN model.

| | | Epinions | | | | DBLP | | | | LastFM | | | |
|---|---|---|---|---|---|---|---|---|---|---|---|---|---|
| | | Internal.↑ | Cross.↑ | Overall↑ | Bias↓ | Internal.↑ | Cross.↑ | Overall↑ | Bias↓ | Internal.↑ | Cross.↑ | Overall↑ | Bias↓ |
| SAGE | Orig. | 31.68 | 28.91 | 30.69 | 2.77 | 69.27 | 14.62 | 56.41 | 54.65 | 32.84 | 14.84 | 26.80 | 18.00 |
| | Debias | **36.98** | **34.45** | **36.08** | **1.63** | **77.18** | **31.54** | **66.23** | **45.64** | 32.73 | **15.13** | **26.80** | **17.60** |
| GIN | Orig. | 33.49 | 30.97 | 32.59 | 2.52 | 56.66 | 16.86 | 47.29 | 39.80 | 33.63 | 16.91 | 28.00 | 16.72 |
| | Debias | **40.56** | **39.39** | **40.20** | **1.26** | **60.61** | **34.03** | **54.35** | **26.58** | 32.13 | **19.88** | **28.00** | **12.25** |
| GAT | Orig. | 39.30 | 34.90 | 37.73 | 4.40 | 66.25 | 22.47 | 55.94 | 43.78 | 32.28 | 12.76 | 25.70 | 19.52 |
| | Debias | **39.58** | **36.13** | **38.35** | **3.45** | **75.66** | **37.46** | **66.66** | **38.20** | **34.24** | **16.32** | **28.20** | **17.92** |
| PPRGo | Orig. | 42.86 | 28.75 | 37.83 | 14.11 | 85.71 | 41.14 | 75.28 | 44.58 | 34.84 | 17.51 | 29.00 | 17.33 |
| | Debias | **47.44** | **42.34** | **45.62** | **5.10** | **90.54** | **55.64** | **82.32** | **34.90** | **35.14** | **19.88** | **30.00** | **15.26** |
| LightGCN | Orig. | 46.43 | 37.11 | 43.11 | 9.32 | 85.95 | 47.41 | 76.88 | 38.54 | 38.16 | 16.62 | 30.90 | 21.54 |
| | Debias | **51.49** | **45.31** | **49.29** | **6.18** | **93.55** | **65.33** | **86.90** | **28.22** | **38.31** | **17.51** | **31.30** | **20.80** |
| UltraGCN | Orig. | 30.62 | 5.81 | 21.78 | 24.81 | 95.74 | 63.82 | 88.22 | 31.92 | 32.73 | 15.73 | 27.10 | 17.00 |
| | Debias | **44.13** | **43.66** | **43.96** | **0.67** | **97.14** | **71.71** | **91.15** | **25.43** | **35.29** | **18.40** | **29.60** | **16.89** |

- After proposing our method, all base models get improved on both internal-links and cross-links in most cases, which results in an improvement in the overall performance. The reasons for this impressive improvement are two-fold. For one thing, supervision augmentation directly helps boost the performance of cross-links. For another, the augmented supervisions play a regularization role in the learning of node embeddings, thus, the overall quality of representations can be improved.

- The increase of cross-links is much higher than that of internal-links, and this contributes to the decrease of bias. For instance, when our framework is applied to LightGCN on DBLP, the performance of internal-links improves from 85.95% to 93.55%, and the performance of cross-links improves from 47.41% to 65.33%, thus the bias decreases from 38.54% to 28.22%.

## 4.3 Ablation Study

In this section, we want to explore: 1) the impact of supervision augmentation on our framework; 2) the impact of the embedding fusion module on our framework; 3) the impact of two auxiliary loss functions in Eq.(4); 4) the impact of dynamic training strategy during training embedding fusion module. As an example, we show the results of LightGCN on two datasets in Table 3. All results are based on Jaccard-based augmentation.

We first replace the augmented cross-links signals, which are selected through

Table 3: Ablation studies with LightGCN as the base model. The results (Hits@50) are reported in percentage (%). The best results are **bold**, and the runner-up is underlined.

| | Epinions | | | DBLP | | |
|---|---|---|---|---|---|---|
| | Internal.↑ | Cross.↑ | Overall↑ | Internal.↑ | Cross.↑ | Overall↑ |
| Original | 46.43 | 37.11 | 43.11 | 85.95 | 47.41 | 76.88 |
| Ours | **51.49** | 45.31 | **49.29** | **93.55** | 65.33 | **86.90** |
| - Augment | 49.43 | 40.97 | 46.77 | 92.76 | 60.97 | 85.24 |
| - Fusion | 44.31 | **46.39** | 44.87 | 72.92 | **66.97** | 71.53 |
| - Auxiliary Loss | 49.63 | 41.55 | 46.75 | 85.31 | 60.61 | 84.85 |
| - Dynamic | 48.63 | 41.58 | 46.12 | 93.09 | 59.08 | 85.08 |

Jaccard coefficient, with random node pairs across different communities, and denote this variant as "-Augment". There are two main observations: 1) After adding random-augmented signals, the overall performance still shows a significant improvement compared to the base model, which indicates that the embedding fusion module can effectively filter out useful information from the augmented embeddings and fuse it into the original embeddings; 2) Compared with our framework, the random-augmented framework demonstrates unsatisfactory results on reducing the performance disparity between cross-links and internal-links, such as slightly reducing this kind of bias on Epinions dataset from 9.32% (46.43% - 37.11%) to 8.46% (49.43% - 40.97%), and this indicates the importance of our supervision augmentation method on debias.

Secondly, after removing the embedding fusion module (denoted as "-Fusion"), although the cross-links performance of LightGCN gets significant improvement, i.e., from 37.11% to 46.39% on the Epinions dataset and from 47.41% to 66.97% on the DBLP dataset, the internal-links performance and overall performance get severe influence comparing with our framework. Especially on DBLP, the internal-links performance of LightGCN is deceased by 13.03% (from 85.95% to 72.92%), while our framework could even slightly improve the performance of internal-links. It indicates that, after proposing the embedding fusion module, our framework could effectively filter out noisy augmented signals to avoid performance deterioration.

Finally, after discarding the two auxiliary loss functions of twin GNNs in Eq.( 4) ("-Auxiliary Loss") or setting a fixed learning rate $\gamma^{F*}$ and training step $S^*$ in the embedding fusion module ("-Dynamic"), all metrics deteriorate slightly compared to our proposed framework. This matches our expectation, because both the lack of guidance from auxiliary loss and a fixed training strategy would inevitably influence the quality of two sets of input embeddings, thus influencing the final embeddings.

## 4.4 Comparison with Other Competitors

In this section, we aim to compare our methods with several powerful competitors on reducing the bias between internal-links and cross-links. Specifically, we choose five highly related methods as our baselines, including FairWalk [30], CFC [4], FairAdj [20], FLIP [28], and UGE [38], and we replace the required sensitive attributes in these methods with community memberships detected by Louvain algorithm [2] in advance. To

Table 4: Comparison with several competitors on both utility and debias. The average results (in Hits@50) are reported in percentage (%) after repeating each method three times. The best results are **bold**, and the runner-up is underlined.

|  | Epinions | | DBLP | | LastFM | |
|---|---|---|---|---|---|---|
|  | Overall↑ | Bias↓ | Overall↑ | Bias↓ | Overall↑ | Bias↓ |
| FairWalk | 23.75 ± 1.1 | 7.16 ± 0.7 | 48.81 ± 1.3 | 50.00 ± 2.1 | 23.05 ± 0.6 | 21.78 ± 1.0 |
| CFC | 27.35 ± 0.4 | 2.01 ± 0.3 | 54.12 ± 2.1 | 49.37 ± 1.6 | 24.28 ± 0.5 | 17.79 ± 0.3 |
| FairAdj | – | – | – | – | 23.97 ± 1.2 | 19.05 ± 0.9 |
| FLIP | 25.65 ± 0.6 | 4.65 ± 0.5 | 55.23 ± 1.3 | 45.93 ± 1.1 | 22.73 ± 0.4 | 18.23 ± 0.5 |
| UGE | 27.22 ± 0.9 | **1.20** ± 0.2 | 54.66 ± 1.8 | 54.36 ± 2.3 | 25.50 ± 0.9 | **14.00** ± 0.7 |
| GraphSAGE | 30.69 ± 0.9 | 2.77 ± 0.3 | 56.41 ± 1.5 | 54.65 ± 1.6 | 26.80 ± 0.5 | 18.00 ± 0.6 |
| Debias-SAGE | **36.08** ± 0.7 | 1.63 ± 0.2 | **66.23** ± 1.9 | **45.64** ± 1.7 | **26.80** ± 0.4 | 17.60 ± 0.3 |

be fair, all models' embedding dimensions are set to 64, and other details on hyper-parameters are listed in Section B in the Appendix for saving space. Note that, due to the algorithm's requirement on the multiplication of the adjacency matrix, FairAdj reports "OOM" errors when deployed on Epinions and DBLP. For comparison, we report the results of GraphSAGE [13] trained with our framework (denoted as Debias-SAGE). The same as Section 4.2, we take Jaccard-based augmentation on Epinions and DBLP and take random walk-based augmentation on LastFM.

As is shown in Table 4, our method achieves state-of-the-art results on overall performance with top-ranking debias results. Especially on the DBLP dataset, our method's overall performance far exceeds that of all other baselines, while achieving the best results in reducing the bias between internal-links and cross-links. We believe the reasons are two-fold: 1) With the model-agnostic design of our framework, we could easily deploy our method on many powerful GNN models. 2) Instead of adding extra constraints in the loss function or modifying the objective functions, our method aims at addressing the bias between internal-links and cross-links from a data perspective, and the supervision augmentation plays a regularization role in boosting the overall performance.

## 4.5 Empirical Findings on Easing Information Cocoons

With emphasizing the performance on cross-links, one merit of our model is the potential capability to ease information cocoons, which can be empirically verified from the following two aspects.

**Debiased Recommendation.** In this part, we further explicitly reveal whether our framework can provide debiased recommendations for nodes to ease the information cocoons. In detail, we train several powerful GNN models, including Light-GCN [14], PPRGo [3], and UltraGCN [27] on Epinions and DBLP. Next, we randomly select 2000 source nodes and observe the average proportion of internal-links in their historical data and their recommendation lists given by GNNs. As is shown in Figure 4, the recommendation lists given by GNN models significantly magnify the nodes' original preference on internal-links. From the perspective of the users (or nodes receiving recommendations), such biased recommendations will distort users' true preference by ignoring their niche interests (cross-links) while emphasizing the mainstream interests (internal-links), and make users feel confined to limited domains. For comparison, we provide the results of our enhanced GNN models as well, and it can be seen that after proposing our method (denoted as "Ours"), the distributions of

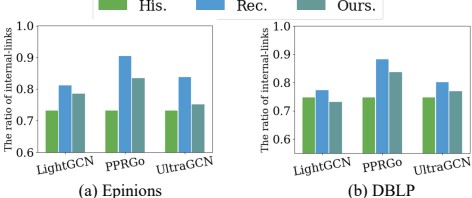

Figure 4: The internal-links proportion of historical distribution and multiple GNNs' recommendations before and after applying our method. "His.", "Rec.", and "Ours." denote the historical interactions, original recommendations, and our debiased recommendations, respectively.

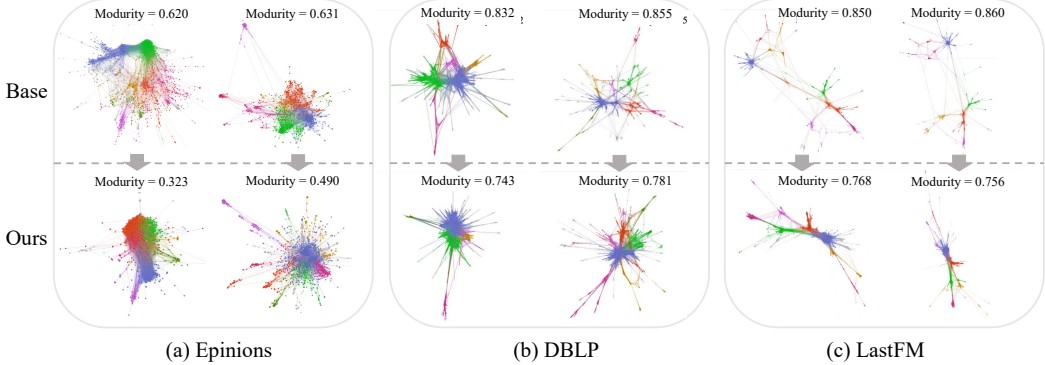

|  | (a) Epinions | (b) DBLP | (c) LastFM |

Figure 5: The visualization of subgraphs reconstructed by the embeddings learned through the base model (above) and ours (below). Note that, each column shows the visualization of the same subgraph, and subgraphs generated by our approach show minor topological isolation phenomenons.

recommendations become closer to that of historical data, which indicates that our approach could effectively address GNN models' biased recommendation and thus ease the information cocoons.

**Visualization of Breaking Filter Bubbles.** To visualize our proposed framework's effect on alleviating information cocoons, we use Gephi[6] to show the graphs reconstructed by node embeddings learned through the base model and our model, respectively. To be specific, we take UltraGCN [27] as the base model here and only a subset of nodes from different communities are shown for conciseness. Note that, all the hyper-parameters in the Gephi platform are set to be the same, and the visualization results reflect the natural layout of the reconstructed networks. The visualization results are shown in Figure 5, and node colors represent the community tags. To better understand the visualization results, we also provide the modularity value in each subgraph, which measures the strength of the division of a network. It can be seen that after proposing our framework, except for some extreme outliers, the connections between most communities are becoming closer, and the isolation between communities is broken down. The reduced modularity values can also support our claims. These results indicate that the embedding generated by our model can effectively facilitate the generation of cross-links in a network, thus alleviating information cocoons.

## 5   Conclusion

In this paper, we aim to address the bias in GNN-based link prediction, especially the bias based on cross-links (links cross communities) after considering its specific roles in easing information cocoons and connecting different communities. We further break the paradigm of modifying objective functions in prior works while pursuing debiased performance, and turn to address the bias issue on cross-links from a data perspective, which could effectively alleviate the bias by even boosting models' prediction utility. Specifically, by borrowing the concept in retrieval-based learning paradigms, debiased node embeddings are generated as demonstrations. An embedding fusion module with dynamic training strategies is also proposed to ensure that the final embeddings could retain the merits of both original embeddings and debiased embeddings. Experiments on three real-world datasets with six base GNNs indicate that our framework could not only reduce the performance disparity between internal-links and cross-links but also significantly improve the overall performance, and further alleviate the potential information cocoons.

**Limitations & Discussions.** Firstly, although we have greatly reduced the bias between internal-links and cross-links, the experimental results indicate that the bias is not clearly eliminated, which implies that data bias may not be the only reason. What's more, despite the supervision augmentation having a potential regularization effect, and leading to improved performance on internal-links, we currently lack theoretical underpinnings and rigorous analysis for this phenomenon. However, we find some potential connections between our work and counterfactual learning. In our settings, supervision augmentation will introduce plenty of unobserved/counterfactual samples for GNN learning. This kind of counterfactual learning may help GNN to better capture the intrinsic relationships between a pair of nodes, thus giving a more accurate prediction.

---

[6]https://gephi.org/

## Acknowledgement

The work is supported by the National Natural Science Foundation of China (No.62172174, No.61932004).

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

# A    Further Analysis on the Role of Cross-links

## A.1    The relationship between cross-links and information cocoons

To fully understand the relationship between cross-links and information cocoons, we conduct the following experiments for analysis.

● **Experimental settings.** Based on the communities detected by Louvain algorithm [2] in advance, we get the internal-links and cross-links of a network, and here we take Epinions and DBLP, two real-world social networks as examples. The detailed dataset information is described in Section 4.1. Next, we borrow the concept of message propagation from the Friedkin-Johnsen dynamics model [6] and revise its formula to simulate the information propagation with randomly initialized node embeddings:

$$\mathbf{Z}_i^t = \frac{\mathbf{Z}_i^{t-1} + \sum_{j \in \mathcal{N}_i} w_{ij} \mathbf{Z}_j^{t-1}}{|\mathcal{N}_i| + 1} \tag{9}$$

where $\mathbf{Z}_i^t$ denotes the embedding of node $i$ at iteration $t$, and $\mathcal{N}_i$ represent the neighbors of node $i$. $w_{ij}$ is a manually controllable reweight scaler determined by the type of link $\langle i, j \rangle$. At each iteration, each node will update its embedding with the weighted average embeddings from its neighbors and itself. For simulating the lack of cross-links, we weaken the role of cross-links in information propagation by tuning the $w$ for cross-links from 1 to 0, and setting $w$ for internal-links to 1.

We further use *Calinski-Harabasz* (CH) index [5, 10] to measure the extent of the information cocoons phenomenon in a network, which can be calculated as follows:

$$CH_C = \frac{SSB_C(\mathbf{Z}^t)}{SSW_C(\mathbf{Z}^t)} \cdot \frac{N - C}{C - 1} \tag{10}$$

where $SSW_C$ and $SSB_C$ are functions to measure the within-cluster dispersion and between-cluster dispersion, respectively [10]. $N$ denotes the number of nodes, and $C$ denotes the number of communities. A higher CH index score indicates that node embeddings are more polarized among communities, which further illustrates the extent of the information cocoon problem in the current network.

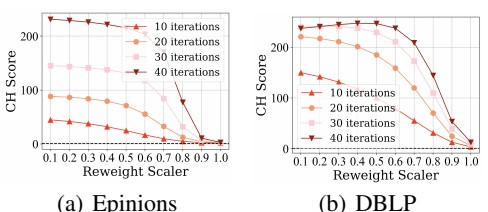

(a) Epinions          (b) DBLP

Figure C1: The distribution of CH index score wrt. the weight of cross-links during propagation. A higher value indicates more severe information cocoons. The dashed lines indicate the CH index score under a normal setting.

● **Experimental results and analysis.** In Figure C1 we show the CH index scores with node embeddings at different propagation iteration $t$, and we can observe that, as the information propagation weight $w$ for cross-links decreases, the CH index score increases consistently and far exceeds that in normal settings ($w = 1$), which indicates that the final node embeddings present more serious polarization problems among communities. Since the information in a single community is relatively limited as shown in Figure 2, the information cocoon problem actually becomes more severe with the lack of cross-links.

## A.2    The relationship between cross-links and graph conectivity

By borrowing the concept of network diffusion, we try to explore the role of cross-links in graph connectivity in this part. Specifically, we apply a classic model in network diffusion: the SI model [18], to simulate the process of information propagation. In this model, each node is randomly initialized with a status called *susceptible* or *infected* at the beginning. During the diffusion iteration process, SI assumes that each infected node could infect its susceptible neighbors with probability $p$, and once a node becomes infected, it stays infected until the end of network diffusion, i.e. there are no more new infected nodes in a new iteration.

In order to provide a clear and vivid illustration, we take one of the most representative social networks – Zachary's karate club[7] as an example. All nodes are divided into four non-overlapping

---
[7] https://en.wikipedia.org/wiki/Zachary%27s_karate_club

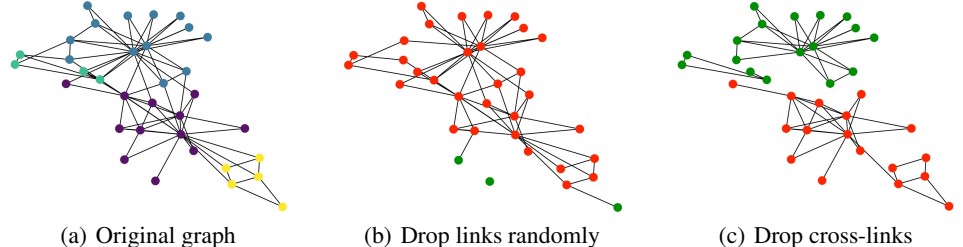

| (a) Original graph | (b) Drop links randomly | (c) Drop cross-links |

Figure C2: The visualization of diffusion simulation with SI model. Red nodes denote *infected* nodes, and green nodes represent *susceptible* nodes. (a) The original graph of the karate club. Nodes with the same color denote a community. (b) Infected graph after dropping some random edges. (c) Infected graphs after dropping some cross-links.

communities by Louvain algorithm [2] in advance, and node #0, which stands for the instructor in this club, is initialized as the only infected node at iteration 0. We further randomly remove 80% cross-links in the graph before starting the simulation. For getting a more convincing conclusion, the network diffusion process on a graph with the same number of random edges removed is also simulated for comparison.

The final visualization results are shown in Figure C2. It can be seen that although we remove some edges randomly from the whole graph, the infected node #0 still propagates its information to almost all nodes in the club (red nodes in the figure) successfully. In contrast, after removing the same number of cross-links, there appears to be an obvious information isolation phenomenon, and nearly half of the nodes remain *susceptible*. In other words, cross-links plays a role in bridging two different communities during network diffusion, and it would be hard for a node to send or receive messages from other communities without enough cross-links. In this way, the existence of cross-links plays a key role in preserving graph connectivity.

## B    Additional Experimetal Settings

### B.1    Base Models and Baselines

Here we introduce additional details for the base models and baselines used in our experiments.

(1) Base models

- **GraphSAGE** [13]: GraphSAGE is an inductive learning framework for generating node embeddings, which samples a fixed number of neighbors during aggregation to alleviate the "neighborhood explosion" issues.

- **GIN** [41]: GIN is a graph neural network that is theoretically as powerful as the Weisfeiler-Lehman test with injective aggregation, combination, and readout functions.

- **GAT** [37]: GAT deploys the attention mechanism during aggregation to capture the neighborhood information with different weights.

- **PPRGo**[3]: By utilizing a Personalized Page Rank matrix to approximate the propagation and aggregation steps with multi-layer graph convolution, PPRGo greatly improves the efficiency and effectiveness on large graphs.

- **LightGCN** [14]: LightGCN empirically finds the redundancy of feature transformation and non-linear activation functions, and greatly simplifies the model architecture with even higher performance on the recommendation tasks.

- **UltraGCN** [27]: Based on LightGCN, UltraGCN further theoretically simplifies the model architecture with approximating the infinite-layer information propagation and aggregation.

(2) Baselines

- **FairWalk** [30]: Instead of a random walk in node2vec, FairWalk chooses its next hop by considering the sensitive attributes in the neighborhood, which successfully mitigates the unfairness related to the sensitive attribute.
- **CFC** [4]: To ensure that the learned embeddings are not correlated with sensitive attributes, such as age or gender, CFC introduces an adversarial framework to enforce fairness constraints.
- **FairAdj** [20]: With learning to assign each edge with different weights, FairAdj generates a fair adjacency matrix and greatly improves the dyadic fairness with comparable utility in link prediction tasks.
- **FLIP** [28]: Concentrating on bursting the filter bubbles in social networks from a dyadic fairness perspective, FLIP also utilizes an adversarial learning framework to generate non-sensitive node embeddings for further link prediction.
- **UGE** [38]: UGE aims at learning unbiased graph embeddings from an unobserved graph, which involves no sensitive information, and further derives three kinds of variants namely UGE-w, UGE-r, and UGE-c.

## B.2 Evaluation Metric

In this work, we apply Hits@50, which is widely adopted in other research [43, 45] and OGB leaderboard [15], as our main evaluation metric to measure the link prediction performance of different GNN models. The Hits@50 can be computed by:

$$Hits@50 = \frac{1}{N_{test}} \sum_{i=1}^{N_{test}} \mathbb{I}(rank_i < 50) \tag{11}$$

where $N_{test}$ represents the sample size of test set, and $\mathbb{I}$ represents an indicator function. $rank_i$ denotes the similarity ranking of the $i$th sample.

## B.3 Reproducibility

**Dataset.** For each dataset, we randomly sample and remove 5% of links in the original graph to construct the validation set and test set, and the remaining links are treated as the training set. Each true sample will be ranked among a set of 100000 randomly sampled negative links for evaluation[8]. Note that, there is no side information, such as node features or link attributes, involved during our experiments, and we assign each node on the graph with a learnable embedding vector for training.

**Hyper-parameters.** As a model-agnostic framework, we deploy six kinds of GNN models as backbones, including GraphSAGE [13], GIN [41], GAT [37], PPRGo [3], LightGCN [14], and UltraGCN [27]. For all these models, we set the output embedding dimension as 64. The layer of the embedding fusion module is set to 1. The learning rates for twin GNNs are both set as 0.001 after grid search. As for the hyper-parameters in Eq.(8), $\alpha$ is set to be 0.005, and $T$ is selected from $\{10, 25, 50\}$ depending on the datasets and base models, and $\beta$ is set to be 20. Augmentation ratio $k$ is searched from $\{0.75, 1.0, 1.25\}$ for each dataset. Both weight decay and dropout rate are set to 0.

In particular, for GraphSAGE, we adopt a mean-pooling during aggregation; for GIN, we apply a linear layer to update node features and use max-pooling during aggregation; for GAT, we use 4 attention heads in each layer; for PPRGo, we set $\alpha$ as 0.3, the walk length as 100; for LightGCN, we set the layer number as 2 and use the final layer's output as embeddings; for UltraGCN, we set the number of negative samples as 64, $\lambda$ as 0.8, $\gamma$ as 3.5.

For Fairwalk, we follow the settings in the original paper and set the walk number to 20, and the window length to 80; for CFC, we set the training steps of the discriminator as 5; for FairAdj, we set $T_2$ to 15 and $\lambda$ to 10; for FLIP, we take the suggestions in the original implementation, and the settings are $\alpha(0.1)$, $\beta(0.2)$; for UGE, we deploy the weighting-based variant as our baseline given that there is no non-sensitive attribute in our settings.

## B.4 Details on LastFM Dataset

Due to the heterogeneity of the recommendation datasets, it is hard to directly deploy community detection algorithms on the original networks and define the corresponding internal-links and cross-links.

---

[8]Here we follow the evaluation protocol in OGB [15], which is widely used in research.

To this end, inspired by ItemCF [22, 31], we first generate an item-item graph according to the co-occurrence relationship. For example, given a pair of items $< v_1, v_2 >$, if they both have interactions with user $u$ at least $O$ times, where $O$ is a hyperparameter to control the confidence of generated item-item graph, there will form a link between $v_1$ and $v_2$. Next, we can deploy our framework on the generated item-item graph for learning debiased item embeddings $\mathbf{I} \in \mathbb{R}^{|V| \times D}$, where $|V|$ represents the number of items and $D$ represents the embedding dimension. After that, item similarity matrix $\mathbf{I}^2 \in \mathbb{R}^{|V| \times |V|}$ is calculated, which is further used for the final recommendation:

Table C1: The comparison between our implementations and normal implementations on LastFM. The average results are reported after repeating each method five times.

| | | LastFM (Hits@50) |
|---|---|---|
| LightGCN | Original | $29.82\% \pm 0.25$ |
| | Ours | $29.30\% \pm 0.21$ |
| UltraGCN | Original | $28.50\% \pm 0.28$ |
| | Ours | $27.32\% \pm 0.19$ |

$$\mathbf{P} = \mathbf{A} \times \mathbf{I}^2 \tag{12}$$

where $\mathbf{P} \in \mathbb{R}^{|U| \times |V|}$ denotes the predicted confidence matrix between users and items, and $\mathbf{A} \in \mathbb{R}^{|U| \times |V|}$ represents the adjacency matrix of the original user-item graph.

In order to prove that our implementation will not affect the performance of the original GNN models, we compare our implementation (denoted as "Ours"), where $O$ is set to be 1, with GNNs trained on the user-item graph normally (denoted as "Original") in Table C1. The results indicate that our implementation does not sacrifice the capability of base GNN models severely to adapt our framework.

## C Further Experiments and Analysis

### C.1 Hyper-parameter Analysis

As the core component in our framework, supervision augmentation plays a key role in mitigating the bias between internal-links and cross-links. To explore its impact in a finer granularity, we vary the augmentation ratio $k$ and see how the performance of our method changes. Specifically, we take LightGCN as the base model and investigate the performance on internal-links (Internal.), cross-links (Cross.), and the whole link set (Overall) by varying $k$ in {0, 0.25, 0.5, 0.75, 1, 1.25}. Without losing generality,

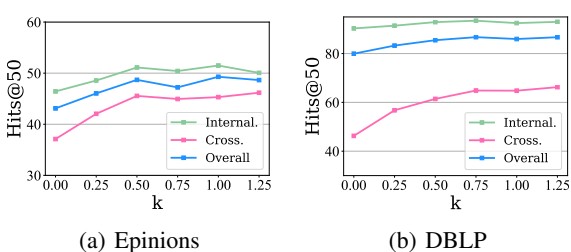

(a) Epinions      (b) DBLP

Figure C3: The impact of augmentation size

here we take Jaccard-based supervision augmentation. As shown in Figure C3, the performance of the two kinds of links increasingly improves as $k$ grows, accompanied by a steady decrease in the difference between them. This is expected because we introduce a large amount of augmented cross-links signals to mitigate the bias. And when $k$ reaches 1, which means $|\mathcal{E}_{in}| \approx |\mathcal{E}_{cr}|$, the framework gradually converges to a stable status. Empirically, a setting $k = 1$ would be a near-optimal option.

### C.2 Alternative Community Detection Algorithm

In this part, we aim to conduct an ablation study on different community detection methods to prove the usefulness of our proposed framework. Since we emphasize the bias from a topological perspective, we prefer to use the community detection algorithm based on graph structure. Specifically, to illustrate the universality of our framework, we conduct experiments based on the METIS [17] algorithm as an ablation study, and the results on three GNNs are shown in Table C2. Specifically, the number of communities is set to be 50 in advance for METIS, and all other hyper-parameters are set to be the same as that in Louvain-based experiments, which can be found in Section B.3. All results are based on Jaccard-based augmentation.

The results indicate that, although we change the community detection algorithm, our framework still successfully mitigates the bias between internal-links and cross-links, and achieves competitive results compared with the Louvain-based results, which verifies our work's compatibility.

Table C2: Ablation study with METIS community detection algorithm on two real-world datasets (Hits@50)

|  |  | Epinions | | | | DBLP | | | |
|---|---|---|---|---|---|---|---|---|---|
|  |  | Internal.↑ | Cross.↑ | Overall↑ | Bias↓ | Internal.↑ | Cross.↑ | Overall↑ | Bias↓ |
| SAGE | Orig. | 36.97 | 19.48 | 30.75 | 17.49 | 69.80 | 19.00 | 54.28 | 50.80 |
|  | Debias | **39.06** | **28.93** | **35.89** | **10.13** | **78.67** | **34.65** | **65.22** | **44.02** |
| GAT | Orig. | 38.33 | 34.77 | 37.15 | 3.56 | 68.62 | 28.15 | 56.26 | 40.47 |
|  | Debias | **39.96** | **36.88** | **38.86** | **3.08** | **75.94** | **42.77** | **65.81** | **33.17** |
| UlltraGCN | Orig. | 27.27 | 11.62 | 20.77 | 15.65 | 97.34 | 70.57 | 89.16 | 26.77 |
|  | Debias | **46.92** | **38.18** | **44.04** | **8.74** | **97.47** | **73.67** | **90.20** | **23.80** |

## C.3 Supervision Augmentation Analysis

In Section 3.3, we design two kinds of data augmentation methods for generating pseudo cross-links supervision signals. Intuitively, if the pseudo supervision signals have a high confidence level, they can provide significant benefits to our framework. To this end, we aim to verify our hypothesis and analyze the impact of different supervision augmentation methods on our framework.

We first statistic the average hop distance between the node pairs generated with different supervision augmentation methods. As shown in Table C3, since we only choose node pairs with the most common neighbors in Jaccard-based augmentation, the hop distance is fixed to 2. When we use random walk-based augmentation, the average distance increases consistently on two datasets, which verifies its effectiveness in covering nodes that are not located in the boundary of communities.

Table C3: The average hop distance between node pairs generated by different supervision augmentation methods

|  | Epinions | DBLP |
|---|---|---|
| Jaccard based | 2.00 | 2.00 |
| Random walk based | 2.69 | 3.14 |

Table C4: Link prediction performance (Hits@50) of internal-links, cross-links, and the whole link set of our methods with random walk based augmentation and corresponding base models on two real-world datasets. The results are reported in percentage (%). We **bold** the results when our framework improves the base GNN model.

Table C4 further presents the performance of our framework with random walk-based augmentation, which is literally described in Section 3.3. Specifically, for providing fair comparison, all hyper-parameters in random walk-based experiments, including augmentation ratio $k$ and others are set to be the same as that in Jaccard-based experiments. Compared to Table 2, it can be shown that the random walk-based framework

|  |  | Epinions | | | | DBLP | | | |
|---|---|---|---|---|---|---|---|---|---|
|  |  | Internal.↑ | Cross.↑ | Overall↑ | Bias↓ | Internal.↑ | Cross.↑ | Overall↑ | Bias↓ |
| SAGE | Orig. | 31.68 | 28.91 | 30.69 | 2.77 | 69.27 | 14.62 | 56.41 | 54.65 |
|  | Fair. | **31.72** | **29.17** | **31.28** | **2.55** | **80.28** | **28.63** | **68.12** | **51.65** |
| GIN | Orig. | 33.49 | 30.97 | 32.59 | 2.52 | 56.66 | 16.86 | 47.29 | 39.80 |
|  | Fair. | **38.35** | **36.89** | **37.12** | **1.46** | **68.12** | **32.07** | **59.64** | **36.05** |
| GAT | Orig. | 39.30 | 34.90 | 37.73 | 4.40 | 66.25 | 22.47 | 55.94 | 43.78 |
|  | Fair. | **40.02** | **36.29** | **37.98** | **3.73** | **75.03** | **32.18** | **64.94** | **42.85** |
| PPRGo | Orig. | 42.86 | 28.75 | 37.83 | 14.11 | 85.71 | 41.14 | 75.28 | 44.58 |
|  | Fair. | **45.36** | **40.12** | **43.49** | **5.24** | **90.48** | **49.51** | **80.47** | **42.08** |
| LightGCN | Orig. | 46.43 | 37.11 | 43.11 | 9.32 | 85.95 | 47.41 | 76.88 | 38.54 |
|  | Fair. | **48.15** | **40.45** | **45.41** | **7.65** | **92.16** | **57.55** | **84.01** | **34.61** |
| UlltraGCN | Orig. | 30.62 | 5.81 | 21.78 | 24.81 | 95.74 | 63.82 | 88.22 | 31.92 |
|  | Fair. | **52.16** | **51.99** | **52.10** | **0.17** | **96.47** | **66.25** | **89.35** | **30.22** |

shows less improvement on cross-links, which results in worse debias results. This observation can be explained by the hop distance in Table C3, which implies that the random walk-based augmentation may have lower confidence due to the longer topological distance between node pairs. However, compared with the base GNNs, the random walk-based framework can still consistently reduce the bias between internal-links and cross-links with improved overall performance.

