# Cross-links Matter for Link Prediction: Rethinking the Debiased GNN from a Data Perspective (Supplemental Materials)

## A Training Algorithm

---
**Algorithm 1** Proposed training process

---
**Input:** Graph $\mathcal{G}$ with link set $\mathcal{E}^O$. Hyper-parameters: $\alpha$, $\beta$, and $T$, learning rate $\gamma^O$, $\gamma^A$.
**Output:** Node embeddings $\mathbf{Z}$
1: Randomly initialize twins GNN models with $\theta^O$, $\theta^A$, embedding fusion module with $\theta^F$.
2: Split $\mathcal{G}$ into $|\mathcal{C}|$ communities and categorize links into internal-links and cross-links.
3: Select augmented supervision signals $\mathcal{E}^A$ with the highest Jaccard coefficient or co-occurrence frequency.
4: **while** not converged **do**
5:     Compute $\mathcal{L}^O$ and $\mathcal{L}^A$ by Eq.(4)
6:     Update twins GNN models: $\theta^O \leftarrow \theta^O + \gamma^O \cdot \nabla_{\theta^O} \mathcal{L}^O, \quad \theta^A \leftarrow \theta^A + \gamma^A \cdot \nabla_{\theta^A} \mathcal{L}^A$
7:     Compute learning rate $\gamma_t^F$ and step size $S_t$ by Eq.(8)
8:     **for** $step = 1$ **to** $S_t$ **do**
9:         Compute $\mathcal{L}^F$ by Eq.(7)
10:         Update embedding fusion module: $\theta^F \leftarrow \theta^F + \gamma_t^F \cdot \nabla_{\theta^F} \mathcal{L}^F$
11:         Update GNN models: $\theta^O \leftarrow \theta^O + \gamma_t^F \cdot \nabla_{\theta^O} \mathcal{L}^F, \quad \theta^A \leftarrow \theta^A + \gamma_t^F \cdot \nabla_{\theta^A} \mathcal{L}^F$
12:     **end for**
13: **end while**
14: **return** $\mathbf{Z}$

---

Here we provide the pseudo codes of our training process, which are the core components helping GNNs to address