# OpenReview forum: "Cross-links Matter for Link Prediction: Rethinking the Debiased GNN from a Data Perspective"
_NeurIPS.cc/2023/Conference — NeurIPS 2023 poster_

### Official Review · Reviewer_Lg87 · 2023-06-11

**Soundness:** 2 fair
**Presentation:** 2 fair
**Contribution:** 2 fair
**Rating:** 6
**Confidence:** 3

**Summary:**

This paper addresses the bias between internal links and cross-links by augmenting cross-links and combining two models consisting of the original and debiased models. Specifically, the authors show that the number of cross-links is fewer than internal links in three real-world datasets. Thus, with Jaccard coefficient score, they augment cross-links and train a debiased model on the augmented graphs. To resolve the trade-off between utility and fairness, they fuse the representation of the original model (which is trained on the original graph), and the debiased model. In experiments, they show that the proposed method resolves the bias and even improves the overall performance.

**Strengths:**

- In contrast with the existing methods, the authors mainly target the bias based on graph topology.
- The suggested method mitigates the bias without compromising performance on several datasets under various architectures.

**Weaknesses:**

- It seems insufficient to demonstrate that internal-links are more common than cross-links based solely on three datasets. For heterophilous graphs, cross-links would be more prevalent when each node label is regarded as a community. In this case, it is more reasonable to augment internal-links, but the suggested method is not able to do it.
- The rationale for supervision augmentation, which is the core component to resolve the bias, seems weak. Why is the Jaccard coefficient score better than other options? Due to its formulation, it would be difficult to augment the connections between a node with a high degree and a node with a low degree. However, using an edge predictor does not have this limitation and is a more simple approach. (Specifically, train an edge predictor and predict edges with this predictor. Then, choose edges based on the confidence of the predictor.)
- The performance of models without fusion is inferior to base models in Table 3. Since supervision augmentation is the core component to mitigate the bias, this component is more important than other components, but it is weak. Based on this observation, it is probable that using UGE as a debiased model would show better performance than the current approach. (Slight tuning is needed to combine UGE.) Then, the contribution of this paper would be marginal.
- Comparing baselines only under GraphSAGE seems insufficient. Could you compare the proposed method with UGE under LightGCN and GAT, too?
- The datasets used in this paper are quite different from the baselines such as FairAdj and UGE. UGE uses  Pokec-z, Pokec-n, and MovieLens-1M, while FairAdj utilizes Oklahoma97, UNC28, Facebook#1684, Cora, Citeseer, and Pubmed. Are there any reasons to evaluate the methods on different datasets? According to FairAdj, Cora, Citeseer, and Pubmed have more internal-links than cross-links. Thus, the performance superiority on these graphs can further support the effectiveness of the proposed method.
- The connections between sentences in the abstract seem unnatural.

Minor issues

- In line 24, “research concerns” seems unnatural. I recommend “research interests”.
- In line 394, “Epilogue” seems non-academic expression. I recommend “Conclusion”.

**Questions:**

- In the ablation study, I’m interested in the experimental details for “-Augment”. I understand it as this policy randomly chooses connections among possible cross-links (Not including internal-links). If not, it would be more persuasive to show the comparison with the policy I described.
- In Table 4, it would be more persuasive to provide the performance of internal-links and cross-links, as shown in Table 2, and 3.
- It is difficult to understand Figure 2. Could you provide a more detailed description of  Figure 2?

**Limitations:**

The authors provide several limitations in the main paper. However, as I mentioned in Weakness, it would be better additionally to address other issues such as the applicability to heterophilous graphs.

---

> ### Author Rebuttal · Authors · 2023-08-09
>
> ### [About the concerns on heterophilous graphs (W1)]
>
> 1. **Additional statistics on multiple heterophilous graphs are provided.** To validate the data bias on heterophilous graphs, we conduct community detection with Louvain algorithm on six datasets that have low homogeneity ratios (Hom. Ratio). Specifically, we follow the definition of homogeneity ratio in [1]: $Hom. Ratio=\frac{\sum_{\langle u, v\rangle\in|\mathcal{E}|}\mathbb{1}(y_u=y_v)}{|\mathcal{E}|}$.
>
>    where $\mathcal{E}$ denotes the edge set,  $y_u$ represents the label of  node $u$ and $\mathbb{1}(\cdot)$ is an indicator function. The numbers of internal-links and cross-links are illustrated in Table R1 within the uploaded PDF. It can be seen that, even in datasets with low homogeneity ratios, cross-links still significantly fall short of internal-links. **This observation further supports that the data bias between internal-links and cross-links is widely existed in the real world, even in heterophilous graphs.**
>
> 2. **Our model's flexibility to augment internal-links.** In fact, the fundamental idea of our approach is to focus on the minority group in the data and then find out high-confidence, unobserved samples for supervision augmentation. **In this way, even if in a context where the number of cross-links exceeds that of internal-links, we can also handle this situation by simply reversing the criteria within the supervision augmentation**, thus augmenting the corresponding minority.
>
> [1] Du et al. GBK-GNN: Gated Bi-Kernel Graph Neural Networks for Modeling Both Homophily and Heterophily. WWW 2022.
>
> ------
>
> ### [About the concerns on supervision augmentation (W2)]
>
> Since we need to train a separate edge predictor for supervision augmentation, we believe that this kind of approach may potentially inherit the data bias between cross-links and internal-links, thereby limiting its effectiveness in mining unobserved cross-links. In contrast, the heuristic methods in our paper require no training process, inherently sidestepping the risk of introducing bias from data. What's more, the experimental results also prove that the methods based on Jaccard coefficient and random walk are simple yet effective.
>
> ------
>
> ### [About the concerns on ablation study and baseline UGE (W3, W4)]
>
> 1. It seems that there might have been a slight misunderstanding regarding the Ablation study.  As a core component in our model, the embedding fusion module is designed to assist GNNs in mitigating the impact of noise introduced by data augmentation. While standalone data augmentation can effectively aid in model debiasing, it may affect the overall performance of the model. In the ablation study, we remove the embedding fusion module and reserve the supervision augmentation, which is denoted as "-fusion". As a result, in Table 3, **"-fusion" achieves the best debias performance -- the lowest Bias**, **while the overall performance is affected.**
>
> 2. **Additional comparisons based on other base GNNs are provided.** As for the comparisons of UGE, we have further implemented UGE on GAT and LightGCN. The results are displayed in Table R4 within the uploaded PDF. As observed, due to the modifications in the objective functions, although UGE can achieve the best debias performance in certain instances, **it is unable to prevent noticeable utility degradation** compared to our methods. **This kind of debias achieved with sacrificing utility is not the admirable goal of our work.**
>
> ------
> ### [About the concerns of datasets (W5)]
>
> We would like to address the reviewer's concerns related to the datasets from the following two aspects:
>
> 1. **The reasons for choosing datasets.** In fact,  **it seems that a universally acknowledged benchmark has not been established yet in the prior work**. To be precise, we statistic the datasets in the prior works in Table R2 within the uploaded PDF, and it can be seen that different methods actually utilize different datasets. Therefore, given the graph scale in the real world, we choose three relatively large datasets from SNAP and RecBole.
>
> 2. **Additional comparisons on the official datasets are provided.** For validating our method's effectiveness, we further conduct experiments and compare the performance of our method with baselines on their official datasets. The results are shown in Table R9 in the uploaded PDF. From the table we can observe that, on the official datasets, **our proposed method still consistently outperforms UGE and FairAdj on both debias and utility**, which further verifies our method's superiority.
>
> ------
>
> ### [About the concerns on presentation and clarification (W6, Q1, Q2, Q3)]
>
> 1. **The details of ablation study "-Augment".** In our "-Augment" setting, we randomly select an equal number of cross-links, without including internal-links in these randomly chosen edges. We will enhance the corresponding descriptions in the final version.
>
> 2. We will consider elaborating on the results of Table 4 in the final version (one additional page is permitted for the accepted paper).
>
> 3. **Additional descriptions about Figure 2.**
>
>    We would like to help you have a comprehensive understanding of Figure 2 by describing the following three aspects:
>
>    - **Experimental settings:** In Figure 2, we initially apply the Louvain algorithm to cluster the item-item graph in the LastFM dataset. Subsequently, for conciseness, we randomly sample 10 clusters and statistic the number of the top 10 commonest labels across the entire graph.
>    - **Observations:** These results are visualized as a heatmap in Figure 2. It can be seen that each community contains its own specific information pattern, i.e. specific label distributions.
>    - **Conclusions:** It's impossible for one single community to encompass all diverse information in a network. This implies the propensity of graphs to form information cocoons with insufficient cross-links as bridges, which further verifies the significance of cross-links.

---

> > ### Comment · Reviewer_Lg87 · 2023-08-12
> > **Change my score to weak accept**
> >
> >
> > I appreciate the detailed response. Most of my concerns have been addressed. Thus, I change my score to weak accept.

---

> > > ### Author Response · Authors · 2023-08-12
> > > **Thank you for the comments**
> > >
> > > Thank you again for your valuable review and reconsideration of scores!

---

### Official Review · Reviewer_Nju4 · 2023-07-05

**Soundness:** 2 fair
**Presentation:** 3 good
**Contribution:** 3 good
**Rating:** 6
**Confidence:** 4

**Summary:**

This work finds that current GNN methods have severe data bias because GNNs like to connect new links inside the local neighbors and ignore the distant ones. To address this problem, the authors investigate the bias across different communities and propose a general framework. In this framework, the authors devise three key components, including supervision augmentation, twins-GNN, and embedding fusion module. To display the effectiveness of either component, the authors perform the ablation study, and with all of them, the Debias model has a promotion on three datasets compared with six GNN backbones.

**Strengths:**

+ The idea of rethinking the data bias, especially the fact that existing GNN models tend to connect local neighbors, is novel and has basic value.
+ The writing is clear and easy to follow.
+ The framework has a strong generality that can be applied to most GNNs (six examples used in this paper).
+ The framework is an end-to-end framework and is easy to accomplish.


**Weaknesses:**

- The clusters should be pre-computed by some community detection algorithms (Louvain algorithm in this main content and METIS algorithm in the appendix); however, the impact of the quality of the community detection is unknown, and the results vary greatly under different community detection algorithms.
- The promotion of link prediction on DBLP and LastFM seems not to be apparent, especially the bias is still very high. (This phenomenon is referred to by the authors in the limitation part.)


**Questions:**

+ This paper uses Jaccard-based augmentation for Epinions and DBLP, and random walk based augmentation for LastfFM because of LastfFM’s high density. Moreover, the authors detail the analysis of two augmentations, but how can we decide which is better before the link prediction experiments?

**Limitations:**

Yes. The authors acknowledge that the data bias might not be the mere reason, and the supervision augmentation lacks theoretical analysis.

---

> ### Author Rebuttal · Authors · 2023-08-09
>
> We are sincerely grateful to the reviewer for the careful evaluation and insightful comments.  We would like to address the concerns raised in the feedback in the following responses.
>
> ------
>
> ### [About the impact of community detection algorithms (W1)]
>
> By comparing the experimental resulst based on Louvain algorithm (Table 2) and Metis algorithm (Table C2 in Appendix D.2), it can be observed that different community detection methods have limited influence on the overall performance of GNN, primarily affecting the debiasing effectiveness. This is mainly due to the sample variations in cross-links and internal-links obtained from different community detection methods. **However, our method consistently demonstrates a certain degree of debiasing capability across different community detection algorithms.** In practical application scenarios, users can define cross-links and internal-links based on their specific requirements, and subsequently employ our approach for training purposes.
>
> ------
>
> ### [About the choice of supervision augmentation (Q1)]
>
> In particular, the choice of supervision augmentation method should primarily be determined by the characteristics of the dataset, such as its density. Additionally, as we highlight when introducing random walk based augmentation, when the communities within the graph are quite extensive, relying solely on Jaccard-based augmentation may inadvertently focus mostly on nodes at the boundaries of communities. Consequently, in such cases, a more effective supervision augmentation strategy would be random walk based methods.
>
> Overall, in this paper, **we have presented two simple yet effective supervision augmentation methods**, both centered around the concept of identifying highly potential cross-links within the graph that have not been observed yet, and we acknowledge the possibility of exploring alternative and potentially more efficient methods in our future work.
>
> ------
>
> ### [About the limited debias performance (W2)]
>
> As we mentioned in the limitation, we observe that the bias between cross-links and internal-links is only mitigated in a limited degree. In this way, we believe that the data bias between internal-links and cross-links may not be the only reason for the poor performance on cross-links, and the performance bias may come from multiple causes such as inbalanced data distributions, GNNs' biased aggregation operations and so on. We sincerely hope that our work can provide a new debias direction for other researchers and we leave the unexplored perspectives as our future work.
>
> ------
>
> We sincerely appreciate the efforts and valuable feedback provided by the reviewer. We genuinely hope that our responses have addressed your concerns and contributed to a better understanding of our research. If you have any further questions or confusion, please do not hesitate to reach out to us. We would be more than willing to assist and provide further clarification.

---

> > ### Comment · Reviewer_Nju4 · 2023-08-13
> > **Response to authors**
> >
> > Thanks for the authors' effort in the response. The authors addressed most of my concerns. I think this work basically has value, but it still needs some further analysis and theoretical results to fulfill the contributions on some of the inherent weaknesses if given a higher score. So, I keep the current score.

---

### Official Review · Reviewer_a1aW · 2023-07-07

**Soundness:** 3 good
**Presentation:** 3 good
**Contribution:** 3 good
**Rating:** 5
**Confidence:** 3

**Summary:**

The authors aim to explore the issue of bias in the link prediction task for GNNs. Specifically, they develop methods to mitigate the bias resulting from graph topology - on internal links versus cross-community links. Their work relies on debiasing node embeddings and a fusion component that retains aspects of both the original and the debiased node embeddings. The main goal is to ensure that the implicit creation of information silos does not degrade link prediction performance. The overall architecture borrow from retrieval model literature and is designed to be partially agnostic to model choice.

**Strengths:**

1. The problem is significant enough in that enough research has been devoted to it in earlier literature, and that topology-induced bias is an interesting direction to consider.

2. The authors experiment on a number of baselines to show the relative superiority of their method. They also include a number of ablation studies to show the effectiveness of their architecture.


3. The architecture is model agnostic and allows for plugging in more powerful GNN models, for example.


4. The loss function of the link prediction objective does not have to be modified (supported by a regularizer) and so the loss surface is not directly affected. Instead, supervised augmentation provides a kind of regularizing effect.


**Weaknesses:**

1. The main weakness of the paper is the small variety of datasets that the experiments have been run on. Ideally there should be multiple kinds of graphs, varying by size (nodes, edges), or even types of communities, their strength or internal cohesiveness and the degree to which they overlap. In comparison, the number of baselines is acceptable. The authors could add more graphs, for e.g. from the SNAP repository and experiment on more graph parameters like the above. Further, synthetic datasets generated by a particular model could help serve as a baseline and also possibly study the evolution of such cross-community bias in social networks.

3. While community detection is done mainly via the Louvain algorithm, one could consider clustering based on node features and other methods such as stochastic block model as another baseline.


**Questions:**

See above

**Limitations:**

The authors address the fact that bias is not entirely eliminated by their procedure and that theoretical support does not yet exist for their contribution. The work has a positive societal impact as the key goal is to reduce bias.

---

> ### Author Rebuttal · Authors · 2023-08-09
>
> Sincere thanks for the reviewer's thorough evaluation and constructive comments. With respect to the reviewer's insightful feedback, we have organized our rebuttal as follows:
>
> ------
>
> ### [About the datasets (W1)]
>
> We would like to address the concerns to the datasets utilized in our paper from the following two aspects:
>
> 1. **The reasons for choosing social networks and recommendation networks.** As we analyzed in Appendix B.1, cross-links are highly essential due to their significance in easing information cocoons such as filter bubbles and echo chambers. These information cocoons widely exist among social networks and recommendation scenarios. In this way, we only utilized these two types of datasets in our work.
>
> 2. **Additional experiments on two real-world datasets are provided.** In fact, our approach can potentially be applied to other types and sizes of datasets as well. In particular, we further investigate the performance of our method on the **Amazon dataset** (from SNAP) and the **Cora dataset**. **The corresponding experimental results based on two kinds of supervision augmentation methods are given in Table R5-R6 in the uploaded PDF.** So far, the datasets we have utilized are summarized in the table below:
>
>    |          | Users   | Items   | Interactions | Type      | Scenario               |
>    | -------- | ------- | ------- | ------------ | --------- | ---------------------- |
>    | Epinions | 75,879  | -       | 508,837      | User-User | Social Network         |
>    | DBLP     | 317,080 | -       | 1,049,866    | User-User | Co-author Network      |
>    | Cora     | -       | 2,078   | 5,278        | Item-Item | Citation Network       |
>    | Amazon   | -       | 334,863 | 925,872      | Item-Item | Co-purchase Network    |
>    | LastFM   | 1,892   | 17,632  | 92,834       | User-Item | Recommendation Network |
>
>    It can be observed that, we conduct experiments on five different kinds of datasets, with graph sizes ranging from 2.1k to 334.8k. The experimental results on all these datasets consistently verify the effectiveness of our model.
>
> 3. **Additional experiments on a synthetic dataset are provided.** We additionally employ the *Stochastic Block Model* (SBM) to generate a synthetic dataset. In particular, the synthetic dataset comprises 4000 nodes and 56128 edges, and we perform experiments using two powerful GNN models: GraphSAGE and GAT. The corresponding results are displayed in Table R7 within the uploaded PDF. **Notably, our approach also consistently outperforms the baseline models in terms of both utility and debias on the synthetic dataset.**
>
>
>
> ------
>
> ### [About the community detection algorithms (W2)]
>
> We agree with the reviewer that the community detection methods are important for our work, and we would like to address the reviewer's concerns from the following two perspectives:
>
> 1. **The reasons for using Louvain algorithms and Metis algorithms.** As we highlight in the introduction, our study primarily concentrates on the link prediction bias based on the graph topology. Consequently, we do not use community detection methods based on node features in our experiments and use two kinds of community detection methods based on graph topology namely Louvain and Metis (mentioned in Appendix D.2).
> 2. **Additional investigations on other community detection algorithms are provided.** To further illustrate the robustness of our approach to community detection methods, in addition to the Metis algorithm mentioned in Appendix D.2, we implement the LPA algorithm for community detection. **The experimental results on LPA are presented in Table R8 in the uploaded PDF**. It can be seen that, although we deploy different community detection method, our method still demonstrate strong capability in easing the bias between cross-links and internal-links with even improved link prediction utility.
>
> ------
>
> We sincerely appreciate the efforts and valuable feedback provided by the reviewer. We genuinely hope that our responses have addressed your concerns and contributed to a better understanding of our research. If you have any further questions or confusion, please do not hesitate to reach out to us. We would be more than willing to assist and provide further clarification.

---

> > ### Comment · Reviewer_a1aW · 2023-08-18
> > **Thanks for the response**
> >
> > Thanks to the authors for the response. I would like to point out that graph structured can be used in SBM style algorithm, e.g., https://www.cs.utexas.edu/users/inderjit/public_papers/kdd_cocluster.pdf. One can run this co-clustering algorithm on the adjancency matrix. Also, one can use node embeddings as node features for co-clustering. I was interested to see how sensitive the method on different choices of community detection algorithms.
> >
> >
> > Summarizing, I think that the paper could include more analysis. I will keep my score as it is.

---

### Official Review · Reviewer_sQxV · 2023-07-19

**Soundness:** 2 fair
**Presentation:** 3 good
**Contribution:** 2 fair
**Rating:** 6
**Confidence:** 3

**Summary:**

The paper introduces a twin-structure framework for mitigating bias in link prediction methods based on Graph Neural Networks. Current link prediction approaches often prioritize performance without considering biases on sensitive attributes of nodes, leading to social risks and information cocoons. The proposed framework divides the graph into communities, distinguishing internal-links from cross-links, and employs supervision augmentation to increase signals for cross-links, generating debiased node embeddings. An embedding fusion module preserves the performance of internal-links while alleviating bias between them and cross-links. Experimental results on real-world datasets demonstrate the framework's effectiveness in reducing bias and improving overall link prediction performance compared to state-of-the-art baselines.

**Strengths:**

* The framework author propose can achieve good improvement on a lot of GNNs.
* The logic of this paper is easy to understand.
* The paper conduct experiments on large datasets.

**Weaknesses:**

* The paper doesn't mention subgraph-based GNN for link prediction.
* The algorithm is not so clear about how to inference.

**Questions:**

* The paper doesn't mention subgraph-based GNN for link prediction. One reason may be those GNNs are not applicable for large graphs and not fast enough for recommendation systems. But I'm just curious does the cross-link matter for those kind of GNNs for link prediction?
* In equation 4 and 7, there seems to be only one negative edge for each positive edge. Why not use some K as hyperparameter? If more than one negative edge per positive one, how will the equations become?
* How to use this framework in the inference step? Do you use augmentation, or just use $Z^O$?
* In Table 3, model without fusion can get best cross-link performance, does that mean cross-links don't matter too much?


**Limitations:**

yes

---

> ### Author Rebuttal · Authors · 2023-08-09
>
> Thanks for your constructive and meticulous comments! We have carefully examined the mentioned issues and have prepared the following rebuttal to address these concerns.
>
> ------
>
> ### [About subgraph-based GNNs (W1, Q1)]
>
> 1. **The reasons for not using subgraph-based GNNs.** Subgraph-based GNNs are usually time-consuming due to excessive subgraph extraction operations, and this issue might be severe when they face large graphs. In light of this, we mainly choose several efficient and effective GNNs as base models. However, we will make sure that more subgraph-based GNNs will be discussed in the final version.
> 2. **The experiments of classical subgraph-based GNN are provided.** To validate if cross-links also matter for subgraph-based GNNs, we opt for a highly classic subgraph-based GNN link prediction model, SEAL [1], for experiments. Specifically, we conduct experiments on the Epinions and DBLP datasets and employed the Louvain algorithm for community detection. The experimental results are presented in Table R3 in the uploaded PDF. It can be seen that, as a classic subgraph-based GNN link prediction model, SEAL also faces significant performance bias between cross-links and internal-links. This observation further verifies the pervasive existence of cross-link bias issues.
>
> [1] Zhang et al. Link Prediction Based on Graph Neural Networks. NeurIPS 2018: 5171-5181.
>
> ------
>
> ### [About the details of inference (W2, Q3)]
>
> Thanks for pointing out this issue. During the Inference stage, we use the node embedding $Z$ for evaluation, which is the output of the embedding fusion module. For better understanding and eliminating potential confusion, **we provide the inference details in Algorithm 1 in the uploaded PDF**.
>
>
>
> ------
>
> ### [About the loss functions (Q2)]
>
> In this work, we have followed the literature [2] and [3] and **employed one of the most classic loss functions in recommendation systems - BPRLoss** as our loss function. Generally, in BPRLoss, there is only one negative sample for each positive sample. However, we concur with the reviewer that incorporating multiple negative samples may positively contribute to improving link prediction performance.  Since the loss function is not the primary focus of our research, we could design other alternative loss functions to handle the situation of multiple negative samples, and we provide two kinds of formulations below:
>
> - $\mathcal{L}=-\sum_{\langle u, v\rangle \in \mathcal{E}^O}\enspace\sum_{\langle u, \hat{v}\rangle \notin \mathcal{E}^O}\enspace\log(\sigma(r_{u,v}-r_{u, \hat{v}}))$ , where $\sigma$ is an activation function like sigmoid.
>
> - $\mathcal{L} = - \sum_{\langle u,v\rangle \in \mathcal{E}^O} \enspace \log \frac{\exp(r_{u,v} / \tau)}{\sum_{\langle u, \hat{v}\rangle \notin \mathcal{E}^O} \enspace \exp(r_{u, \hat{v}} / \tau)}$, where $\tau$ is the temperature parameter.
>
> Both two loss functions aim at maximizing the prediction scores between positive node pairs while minimizing that between negative node pairs. **Note that, all these mentioned loss functions can be easily deployed to our method, which is determined by the practitioners according to their practical applications**. We will make this claim more clear in the final version.
>
> [2] He et al. LightGCN: Simplifying and Powering Graph Convolution Network for Recommendation. SIGIR 2020: 639-648.
>
> [3] Wu et al. Self-supervised Graph Learning for Recommendation. SIGIR 2021: 726-735.
>
> ------
>
> ### [About the concerns on ablation study (Q4)]
>
> It seems that there are some misunderstandings about the "-fusion" in the ablation study and cross-links, and we would like to help you have a better comprehension from the following two key points:
>
> 1. **The role of embedding fusion.** Embedding fusion module is designed to assist GNNs in mitigating the impact of noise introduced by data augmentation. While standalone supervision augmentation can effectively aid in model debiasing, it may affect the overall performance of the model. In the ablation study, we remove the embedding fusion module and **reserve the supervision augmentation, which is denoted as "-fusion"**. As a result, in Table 3, **"-fusion" achieves the best debias performance -- the lowest Bias**, **while the overall performance is affected.**
> 2. **The significance of cross-links.** As demonstrated in Figure 1, the proportion of cross-links in our utilized dataset is relatively small; however, they actually play a significant role in eliminating information cocoons and ensuring graph connectivity as we analyzed in Appendix B.
>
> ------
>
> We sincerely appreciate the reviewer's dedication and insightful comments. We hope that our responses have effectively tackled your concerns and enhanced your comprehension of our study. If you have additional questions or confusion, please feel free to contact us without hesitation. We are more than willing to be engaged in a new discussion and offer assistance.

---

> > ### Comment · Reviewer_sQxV · 2023-08-13
> > **Change my score to weak accept**
> >
> > I appreciate the detailed response. Most of my concerns have been addressed. For the discussion of subgraph-based models, I encourage the authors to include some more recent models which are more efficient (like SUREL+) in the final version. In the end,  I change my score to weak accept.

---

> > > ### Author Response · Authors · 2023-08-13
> > > **Thanks for your comment**
> > >
> > > Thanks for your valuable feedback and reconsideration of scores! We will take your advice and include more recent subgraph-based work in the final version.

---

### Official Review · Reviewer_6Hpo · 2023-07-22

**Soundness:** 2 fair
**Presentation:** 2 fair
**Contribution:** 2 fair
**Rating:** 6
**Confidence:** 3

**Summary:**

This paper addresses the issue of bias in GNN link prediction and proposes a twin-structure framework to mitigate the bias and improve performance. The framework includes an embedding fusion module and a debias module, which work together to reduce the bias between cross-links and internal-links without hurting overall performance. Experiments on three datasets with six different GNNs show that the proposed framework can both alleviate the bias and boost the overall GNN performance.


**Strengths:**

- The paper addresses an important issue of bias in GNN-based link prediction and proposes a novel framework to mitigate the bias.
- The experiment results show that the proposed framework almost always provides both debias and performance gain, even on different GNNs.
- The twin-structure and embedding fusion is simple and clear.
- Limitations are discussed and code is provided for reproducibility


**Weaknesses:**

- Paper presentation can be improved. For example, Figure 5 is too small, numbers are hard to read and colors are hard to distinguish. Also, if space permits, I feel like moving Algorithm 1 in Appendix to the main body would be better.

**Questions:**

1. Any efficiency analysis/convergence analysis/time complexity analysis for better understanding the complexity of the proposed framework compared to standard supervised training.

2. For the mentioned limitation of not having a theoretical analysis. Any ideas about how to approach it? Are there any potential connections to any existing GNN analysis framework?


**Limitations:**

Two limitations are mentioned. 1. The bias is not completely eliminated. 2. Lacking theoretical understanding. I think pointing out these limitations is a plus and both limitations can trigger meaningful future work.

---

> ### Author Rebuttal · Authors · 2023-08-09
>
> We would like to thank the reviewer for these insightful and enlightening comments. Specifically, we aims to address the concerns of the reviewer with the following responses.
>
> ------
>
> ### [About the presentation (W1)]
>
> According to the official guidelines of NeurIPS 2023, accepted papers are allowed to add one extra page for the camera-ready version. In this way, we would increase the figure size and font size in Figure 5, and relocate Algorithm 1 from the Appendix to the main text if our work is accepted at last.
>
> ------
>
> ### [About the efficiency of our model (Q1)]
>
> Thanks for your comment, and we would like to analyze the efficiency of our model through the following two perspectives:
>
> 1. **Time complexity analysis is provided.** Before the time complexity analysis, we would like to give some notations first:
>
>    | Notations          | Descriptions                                            |
>    | ------------------ | ------------------------------------------------------- |
>    | $\mathcal{E}_{in}$ | The set of internal-links in the graph.                 |
>    | $\mathcal{E}_{cr}$ | The set of cross-links in the graph.                    |
>    | $k$                | The augmentation ratio.                                 |
>    | $S_t$              | The training step for training embedding fusion module. |
>
>    As a model-agnostic framework, we assume a specific GNN model requires time *T* to finish the training on a single sample. Then, the time complexity of the base models can be denoted as:
>
>      $T _{base}=(|\mathcal{E} _{in}|+|\mathcal{E} _{cr}|)*T$
>
>    while the time complexity of our method is:
>
>     $\begin{align}T _{our} &=(S _t+2)*(|\mathcal{E} _{in}|+|\mathcal{E} _{cr}|)*T+k(|\mathcal{E} _{in}|-|\mathcal{E} _{cr}|)*T  \\\  &<(S _t+2) * (|\mathcal{E} _{in}| + |\mathcal{E} _{cr}|)*T + k(|\mathcal{E} _{in}|+|\mathcal{E} _{cr}|)*T \\\ &=(S _t + 2 + k)*T _{base} \end{align}$
>
>    where $k$ is the augmentation ratio, which is fixed to 1 or 1.25 in advance. It can be seen that, for each epoch, the upper bound of our algorithm's time complexity is highly dependent on $S_t$. **Fortunately, in our experiments, we deploy a dynamic training strategy, which would set $S_t$ to a relatively small value to avoid unnecessary time expenses at the beginning of training.** To be concrete, for each dataset, the value of $S_t$ starts at 1 at the initial epochs and slowly increases to 20 as the training progresses.
>
> 2. **Convergence analysis is provided.** We further provide the convergence analysis of our models and two base models on Epinions dataset. The results on GAT and LightGCN are illustrated in Figure R1 in the uploaded PDF. From the observation we can find that, **despite the additional time our model requires to achieve convergence, the results demonstrate that we are able to attain comparable performance to the base models within a similar timeframe.** We believe that the main reason for this observation is our dynamic training strategy, which sets a relatively small $S_t$ at the beginning of training, thus greatly reducing the time expenses.
>
> ------
>
> ### [About the theoretical analysis (Q2)]
>
> We believe this is a very inspiring question and we list some existing techniques that may have potential connections with our work.
>
> 1. **Counterfactual learning.** There might be some potential connections between our work and counterfactual learning [1]. In our settings, supervision augmentation will introduce plenty of unobserved/counterfactual samples for GNN learning. To be specific, in this paper, we could regard node pairs as contexts $\mathcal{C}$, structural information like community memberships as the treatment $\mathcal{T}$, and the presence of links as the outcome $\mathcal{O}$.
>
>      The conventional GNN-based link predictor typically follows the formulation $\mathcal{O}=f(\mathcal{C}, \mathcal{T})$. By incorporating counterfactual samples, the objective transforms into finding a GNN model that satisfies $\mathcal{O}=f(\mathcal{C}, \hat{\mathcal{T}})$, where the distribution of treatment has changed. In this manner, optimization guided by these two objectives inherently aids the GNN in more effectively capturing the underlying relationships between node pairs and enhancing its robustness to noise or unessential treatment, thus yielding more accurate predictions.
>
> 2. **Multi-modal fusion.** Multi-modal fusion refers to the process of combining information from multiple distinct sources or modalities to enhance the representation of machine learning. In our work, the twin GNNs actually play two distinct roles in mining graph data, and the embedding fusion may increase the whole model's robustness to noise and variations.
>
> [1] Zhao et al. Learning from Counterfactual Links for Link Prediction. ICML 2022: 26911-26926
>
> ------
>
> We sincerely appreciate the efforts and valuable feedback provided by the reviewer. We genuinely hope that our responses have addressed your concerns and contributed to a better understanding of our research. If you have any further questions or confusion, please do not hesitate to reach out to us. We would be more than willing to assist and provide further clarification.

---

> > ### Comment · Reviewer_6Hpo · 2023-08-13
> > **Response to authors**
> >
> > Thank the authors for answering my questions. My concerns are addressed. The connection to counterfactual learning reads interesting to me. The authors may consider including this part in their final draft if space permits. I will keep my score.

---

### Author Rebuttal · Authors · 2023-08-09

## Response for All Reviewers

We sincerely appreciate the dedication and effort put forth by all the reviewers. We hope that our responses have effectively addressed your concerns and contributed to a deeper understanding of our research. **Due to space limitations, we have included most of the experimental results (figures and tables) in the attached PDF, with corresponding references provided in the rebuttal.** If you have additional questions or confusion, please feel free to contact us without hesitation. We would sincerely like to provide more information and clarification if necessary.

---

### Comment · Area_Chair_ohTV · 2023-08-13
**Thanks for your constructive reviews**

Dear reviewers,

Thanks for your constructive reviews. Looks like most of your reviews are really positive. The authors have submitted their responses to your reviews and answered your questions. Reviewer a1aW and Nju4, could you do us a favor and check if they've addressed your concerns? Appreciate your help!

AC

---

### Decision · Program_Chairs · 2023-09-21

**Decision:**

Accept (poster)

**Comment:**

This paper tackles the issue of bias in GNN link prediction and introduces a twin-structure framework to alleviate the bias and enhance performance. Reviewers concur that the proposed twin-structure and embedding fusion are both straightforward and clear. The experimental results on extensive datasets demonstrate that the proposed framework effectively addresses bias and leads to performance improvements.

However, reviewers have also raised concerns regarding the efficiency of the proposed model, the limited diversity of datasets, the theoretical analysis, certain missing details, and the overall clarity of the paper. The authors' responses have managed to address a portion of the reviewers' concerns, and as a result, some reviewers have subsequently revised their scores. Based on these considerations, I recommend acceptance.